# Multi-Band Array Antenna Sharing a Common Aperture with Heterogeneous Array Elements

Sungsik Wang [1], Hyunsoo Kim [2], Dongyoon Kim [3] and Hosung Choo [4,*]

1   Department of Electrical, Electronic, & Communication Engineering, Hangyang Cyber University, Seoul 04066, Korea
2   Electronic Warfare R & D, LIG Nex1 Co., Sungnam 13488, Korea
3   Radar R & D, LIG Nex1 Co., Yongin 16911, Korea
4   School of Electronic and Electrical Engineering, Hongik University, Seoul 04066, Korea
*   Correspondence: hschoo@hongik.ac.kr

**Abstract:** This paper proposes a multi-band array antenna that shares a common aperture with heterogenous array elements. The multi-band array antenna includes one printed dipole antenna for the S-band and $3 \times 3$ array E-shaped patch antennas for the X-band. The current directions of the printed dipole and E-shaped antenna are orthogonal to each other, which properly diminishes the mutual coupling interference. To decrease the mutual coupling interference among the X-band components, we placed cavities using multiple vias surrounding the X-band components. To check the validity of the proposed design, the unit-cell was expanded to a $12 \times 12$ X-band array configuration, and then the beam steering properties were examined. The proposed antenna's average gains are 5.2 dBi in the S-band and 5.2 dBi in the X-band. The bore-sight gain of the extended array configuration on the ship mast is 35.6 dBi. The results confirm that the proposed design is suitable for MFR applications even in a shared aperture.

**Keywords:** shared-aperture array antenna; multi-function radar; multi-band array; E-shaped antenna; printed dipole antenna

## 1. Introduction

With recent developments in active electronically scanned array (AESA), interest and use of the multi-function radar (MFR) system has been progressively growing. In particular, MFRs that cover both the S-band and X-band are extensively applied in various applications such as surveillance, electronic warfare, missile guidance, and target-tracing [1–6]. In general, the MFR installs separate antenna systems for each function in an enlarged antenna aperture, which significantly increases the radar cross section (RCS) [4–9]. To resolve these problems, the best solution may be to integrate the entire antenna systems into one shared-aperture space. However, the limited space of the shared aperture increases electromagnetic interference due to the decreased distance of the elements. To resolve this mutual coupling effect, various techniques, such as cavity structure [10], meta-surface [11], and electromagnetic band gap structures [12], have been reported. In addition, various attempts in relation to steerable arrays have been made in various fields, such as Ku-band CubeSat applications, 5G Millimeter-wave application, and unmanned aerial vehicle (UAV) applications [13–15]. However, it is difficult to apply in an expandable array radar system in which multiple bands of antennas share a compact aperture. Moreover, in shared-aperture radar antenna, reducing the structural influence of different band antennas is as important as reducing interference between same-band antennas. Although the shared-aperture array system requires more advanced approaches than the existing design method, studies on shared-aperture radar arrays have not yet been sufficiently conducted.

In this paper, we propose a multi-band array antenna that shares a common aperture with heterogenous array elements. The proposed multi-band array comprises one printed

dipole antenna for the S-band and nine E-shaped patch antennas for the X-band. The current directions between the S-band and X-band elements are perpendicular to each other, which diminishes the mutual coupling interference among the different band elements.

To suppress the mutual coupling interference among the X-band components, we placed cavities using multiple vias surrounding the X-band components. To check the effectiveness of the proposed design, the unit-cell was expanded to a $12 \times 12$ X-band array configuration, and it was mounted on a ship mast. The active reflection co-efficients and beam steering properties were then examined in the X-band. The results verify that the proposed design is applicable for MFR applications even in a shared aperture.

## 2. Unit-Cell Design for the Multi-Band Array Antenna

The geometries of the proposed multi-band array antenna are illustrated in Figure 1. The proposed multi-band array antenna is composed of one printed dipole antenna for S-band and nine E-shaped patch antennas for the X-band. They are perpendicularly mounted to each other. The radiating part of the antenna is made of 1 oz copper with a conductivity of $5.87 \times 10^7$ S/m and is printed on dielectric substrates. The RF-35 substrate from Taconic ($\varepsilon_r = 3.5$, tan $\delta = 0.0018$, $d_1 = 0.8$ mm) is used for the S-band printed dipole antenna, and the TLY-5 substrate from Taconic ($\varepsilon_r = 2.2$, tan $\delta = 0.0009$, $d_2 = 3.08$ mm) is used for the X-band E-shaped antenna. To improve the bandwidth property, a coupled-fed structure is employed to the S-band antenna, and two parallel slots are used in the X-band antennas. The S-band antenna contains a folded arm structure for wide beamwidth and reduction of the antenna size. The length of the printed dipole's arm is represented by $w_1$ and $l_1$, and the dipole is fed from the bent strip line on the opposite surface of the substrate. The induced currents of the S-band printed dipole antenna flow in the $z$- and $y$-directions. In contrast, the induced currents of the X-band E-shaped antenna flow in the $x$-direction, perpendicular to the S-band antenna. Since these current directions are orthogonal to each other, the mutual coupling interference can effectively be diminished. Moreover, to decrease the mutual coupling interference among the X-band components, we installed cavities using multiple vias surrounding the X-band components. The geometrical parameters are optimized using the FEKO electromagnetic (EM) simulation software tool [16], as listed in Table 1.

**Table 1.** Design parameters for the proposed array antenna.

| Parameters | Values (mm) | Parameters | Values (mm) |
|:---:|:---:|:---:|:---:|
| $w_0$ | 37 | $l_0$ | 29 |
| $w_1$ | 16.75 | $l_1$ | 14 |
| $w_2$ | 5 | $l_2$ | 9.5 |
| $w_3$ | 6.75 | $l_3$ | 10.5 |
| $w_4$ | 16 | $l_4$ | 15 |
| $w_5$ | 8 | $l_5$ | 9 |
| $w_6$ | 2.5 | $l_6$ | 2.5 |
| $w_7$ | 15 | $l_7$ | 0.8 |
| $w_8$ | 8.3 | $d_1$ | 0.8 |
| $w_9$ | 7.3 | $d_2$ | 3.08 |
| | | $p$ | 2.3 |

To examine the design performance, the multi-band array antenna is implemented and measured in a full anechoic chamber. Figure 2a,b presents the side views of the fabricated printed dipole antenna for the S-band; the S-band element has a folded arm and a coupled-fed structure. Figure 2c illustrates the top view of the fabricated E-shaped antennas for X-band. Extended via cavity structures are employed around the E-shaped antennas in the X-band, as shown in Figure 2c. Figure 2d presents the assembled antenna unit-cell, where the different band layers are firmly combined utilizing a glue-type adhesive. The S-band and X-band antennas are sharing the ground. In addition, the cavity of the X-band antenna and the arms of the S-band antenna are electrically connected, as shown in Figure 2e. Since the unit-cell has one S-band printed dipole and nine X-band antennas, it has totally ten

SMA connectors that are directly connected to the transmitting receiver modules (TRMs) for digital beamforming. Since MFR radar systems using digital beamforming need a shock-stable feeding network, the co-planar waveguide (CPW) feeding network interface with high-durability [17] is considered to be employed in future work.

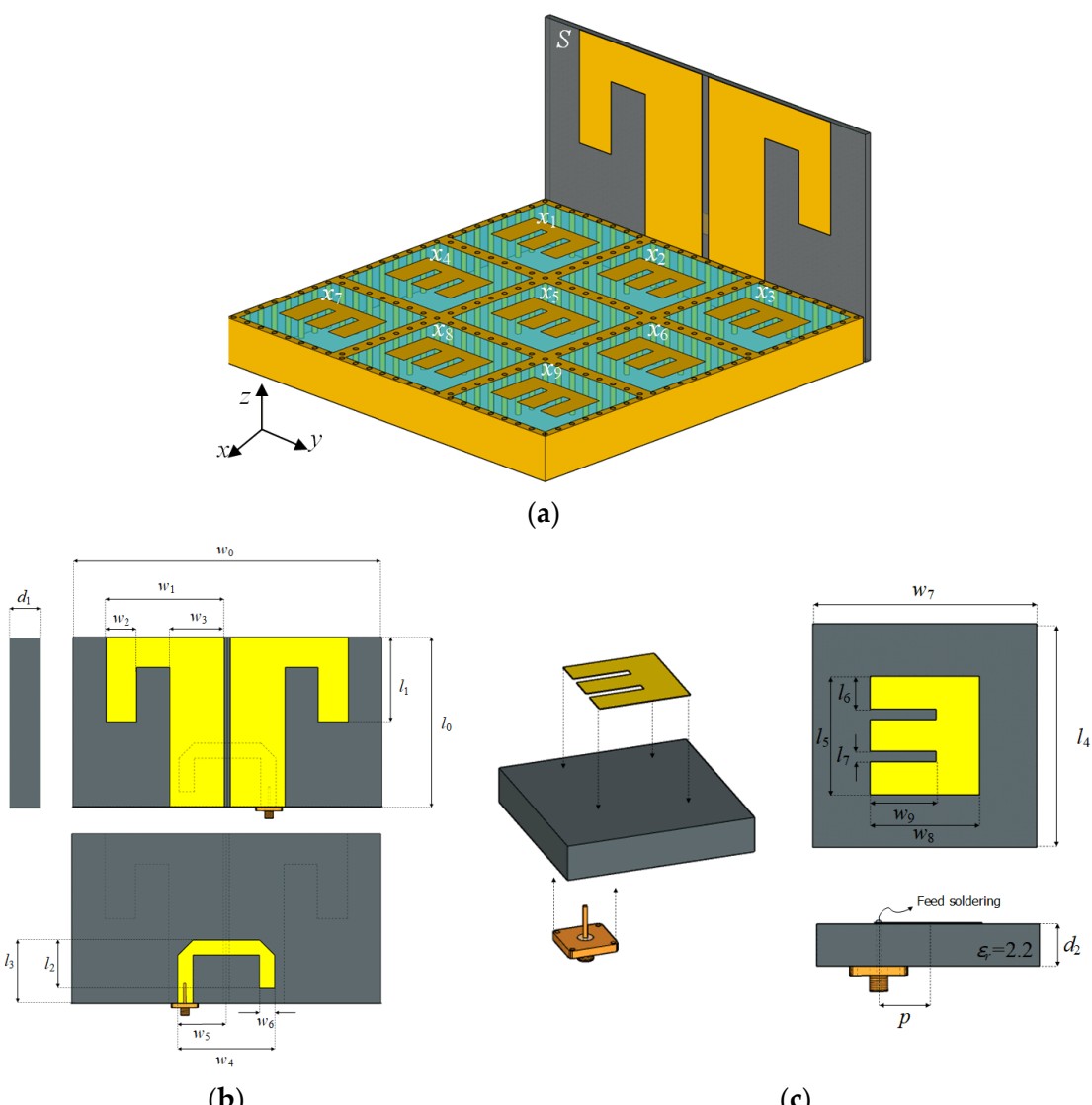

**Figure 1.** The proposed array antenna: (**a**) isometric view; (**b**) printed dipole antenna for S-band; (**c**) E-shaped patch antenna for X-band.

Figure 3 depicts the test setup of the fabricated antenna under test (AUT) with a Styrofoam zig in the full anechoic chamber. The active element patterns and reflections for all the 10 unit-cell elements are completely measured.

Figure 4 represents the bore-sight gains and reflection co-efficients of the implemented proposed antenna. While the solid and dashed line illustrate the measured and simulated reflection co-efficient, the 'x' marks and dotted line denote the measured and simulated bore-sight gains, where the number of the measured points are 10 in S-band and 31 in X-band. The measured reflection co-efficient of the S-band antenna is less than −10 dB from 2.88 GHz to 3.77 GHz, where the bandwidth of 890 MHz is in good accordance with the simulated results, as depicted in Figure 4a. The averaged gains from 2.8 GHz to 3.8 GHz are 5.4 dBi by simulation and 5.2 dBi by measurement. X-band antennas are also measured, and the results show that all X-band antennas have a bandwidth over 1.1 GHz with a reflection less than −10 dB at the target frequency of 9.5 GHz. The measured reflection co-efficient of the central X-band antenna

is less than −10 dB from 8.97 GHz to 10.48 GHz, where the bandwidth is 1.51 GHz (a fractional bandwidth of 15.5%), as shown in Figure 4b. The averaged gains from 8.5 GHz to 11 GHz are 5.8 dBi by simulation and 5.2 dBi by measurement.

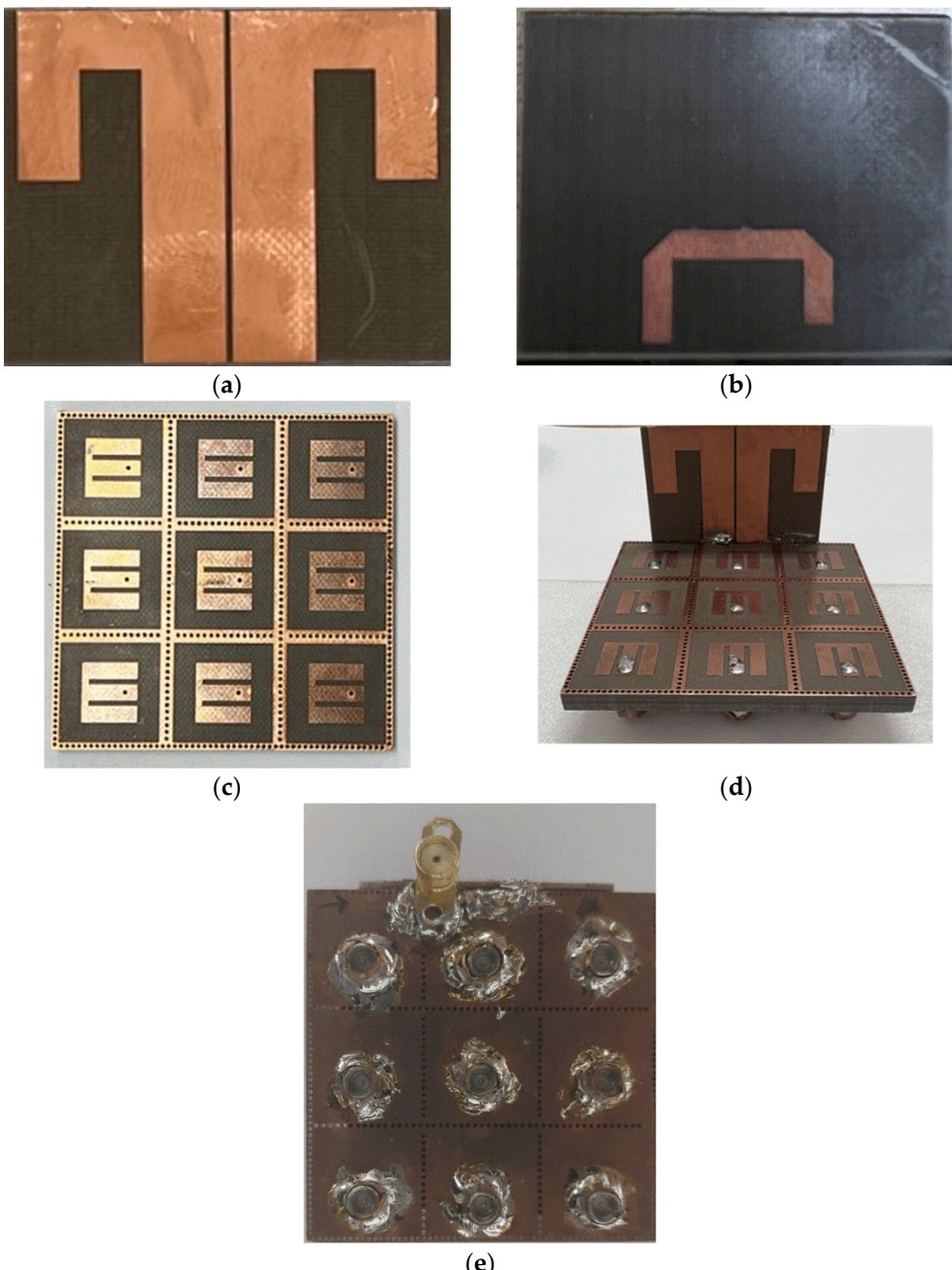

(a)

(b)

(c)

(d)

(e)

**Figure 2.** Photographs of the fabricated multi-band array antenna: (**a**) front view of the S-band antenna; (**b**) back view of the S-band antenna; (**c**) top view of the X-band antenna; (**d**) isometric view of the assembled unit-cell; (**e**) bottom view of the assembled unit-cell.

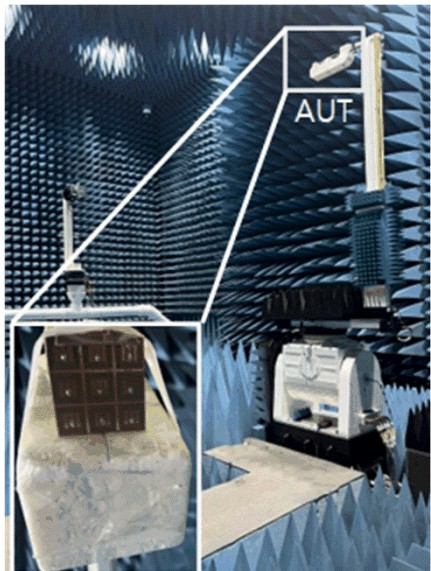

**Figure 3.** Photographs of the test setup in an anechoic chamber for far-field measurements.

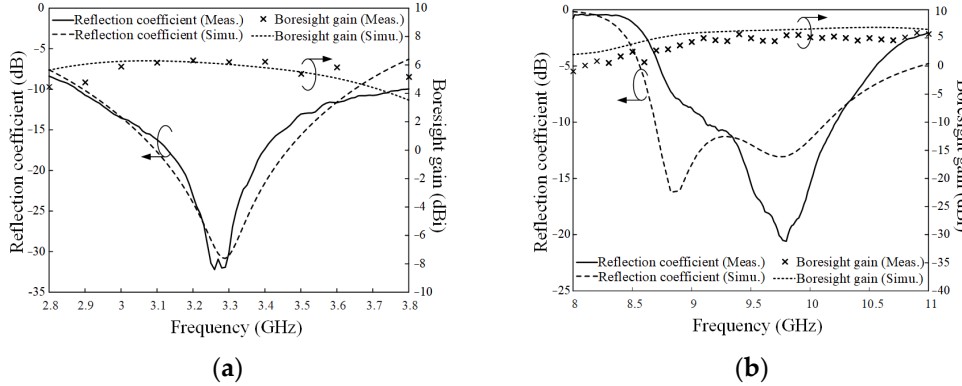

**Figure 4.** The bore-sight gains and reflection co-efficients of the proposed antenna: (**a**) S-band; (**b**) central X-band.

Figure 5 shows the two-dimensional (2D) radiation patterns of the design at 3 GHz and at 9.5 GHz. Figure 5a illustrates the S-band beam pattern, where a half-power beam width (HPBW) for measurement is 82.6°. Meanwhile, Figure 5b presents the central X-band beam pattern, where a half-power beam width (HPBW) for measurement is 61.4°.

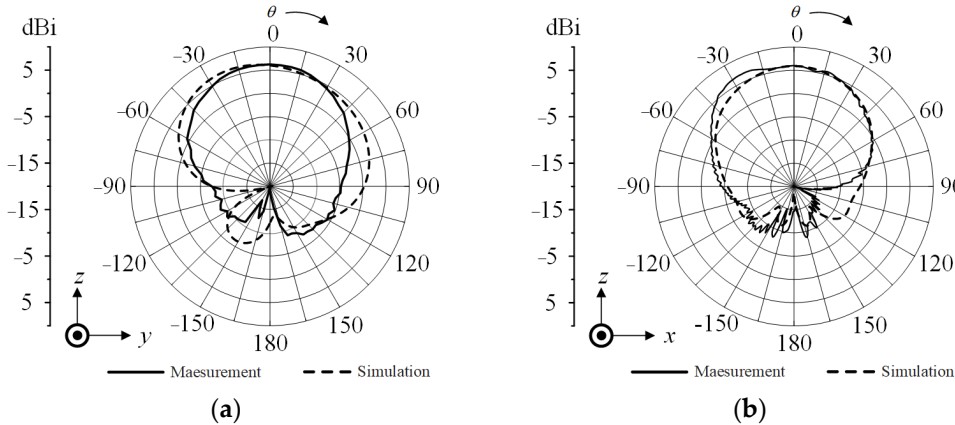

**Figure 5.** 2D radiation patterns: (**a**) at 3.25 GHz; (**b**) at 9.5 GHz.

## 3. Verification of the Proposed Multi-Band Array Antenna

Figure 6 depicts the active reflection co-efficient (ARC) of the central X-band element among the implemented multi-band array antenna. The dashed, solid, and dotted lines present the simulated reflection co-efficient, the measured reflection co-efficient, and the measured ARC (a steering angle of $0°$), respectively. For a fully excited array with a steering angle of $\theta_0$, the ARC of the $m$-th element can be calculated as follows [18,19]:

$$\Gamma_{m.active}(\theta_0) = \frac{\sum_{n=1}^{N} s_{mn}e^{-ju_0(n-1)}}{e^{-ju_0(m-1)}} = \sum_{n=1}^{N} s_{mn}e^{-ju_0(n-m)}, \tag{1}$$

where $d$ is the array distance, and $u_0$ is $kdsin\theta_0$. $S_{mn}$ represents the S-parameter with $m$-th output and $n$-th input in the array. $e^{-ju_0(n-m)}$ is the phase difference between $m$-th port and $n$-th port. The ARC for $0°$ angle steering is under $-10$ dB from 8.5 GHz to 10.2 GHz and is similar to the bandwidth of the reflection co-efficient, as shown in Figure 6a. Figure 6b shows the projected ARCs onto the UV-plane, when the beam steering vectors are weighted to each X-band element [20,21]. The black dashed line indicates the $\pm45°$ steering angle region in the Azimuth-over-Elevation (Az/El) coordinate system [22]. The ARC with the dB scale is expressed as a colored level, where the deeper blue color represents better matching characteristics. The maximum ARC value in the $\pm45°$ steering angle region of the Az/El coordinate system is $-11.7$ dB at 9.5 GHz.

Figure 7 presents the steering beam patterns of the $3 \times 3$ X-band array in the unit-cell at 9.5 GHz. The measured gains with steering angles of $0°$, $15°$, $30°$, and $45°$ are 15.8 dBi, 15.6 dBi, 14.4 dBi, and 12.2 dBi, respectively, whereas the simulated bore-sight gain for a $0°$ steering angle is 14.8 dBi, where the simulated and the measured steering beam patterns are in good accordance, as presented in Figure 7. The beam steering performance is obtained by super positioning the measured active element patterns of all antenna elements [23]. Phase control for each element is conducted by applying the theoretical phase weighting to the measured active element patterns. This assumes that TRMs connected each antenna are capable of ideal phase control. For better beam steering performance, the polarization direction and currents between S-band and X-band antennas are set to be orthogonal to each other. Thus, we did not observe any significant degradation in beam steering performance even with the common aperture configuration. The overall size, application, gain, and SLL of the proposed antenna system are compared with those of reference antennas in Table 2.

**Table 2.** Comparison of the proposed array antenna and reference antennas.

| Ref | Freq. (GHz) | Area | Array | Application | Gain | SLL (dB) |
|---|---|---|---|---|---|---|
| [10] | 21 | $3.85\lambda \times 3.85\lambda$ | $4 \times 4$ | radar and communication | 17.8 dBi | $-16.2$ |
| [11] | 3.5 | $2.3\lambda \times 1.17\lambda$ | $1 \times 4$ | 5G base stations | NA | NA |
| [12] | 5.75 | $0.127\lambda \times 0.127\lambda$ | NA | NA | 1.3 dB increase | NA |
| [13] | 37 | $10.48\lambda \times 10.48\lambda$ | $8 \times 8$ | Ku-band CubeSat | 24.04 dBic | NA |
| [14] | 26.5 | $4.73\lambda \times 0.274\lambda$ | $1 \times 12$ | 5G Millimeter-wave | 14.1 dBi | $-16.2$ |
| [15] | 1.96 | $2.16\lambda \times 0.54\lambda$ | $1 \times 4$ | Unmanned aerial vehicle (UAV) | 9.98 dBi | $-4$ |
| Proposed work | 3.25 | $1.6\lambda \times 1.6\lambda$ | $3 \times 3$ | MFR radar | 15.8 dBi | $-13.3$ |
| | 9.5 | $2.2\lambda \times 2.2\lambda$ | $4 \times 4$ | | 23.24 dBi | $-22$ |

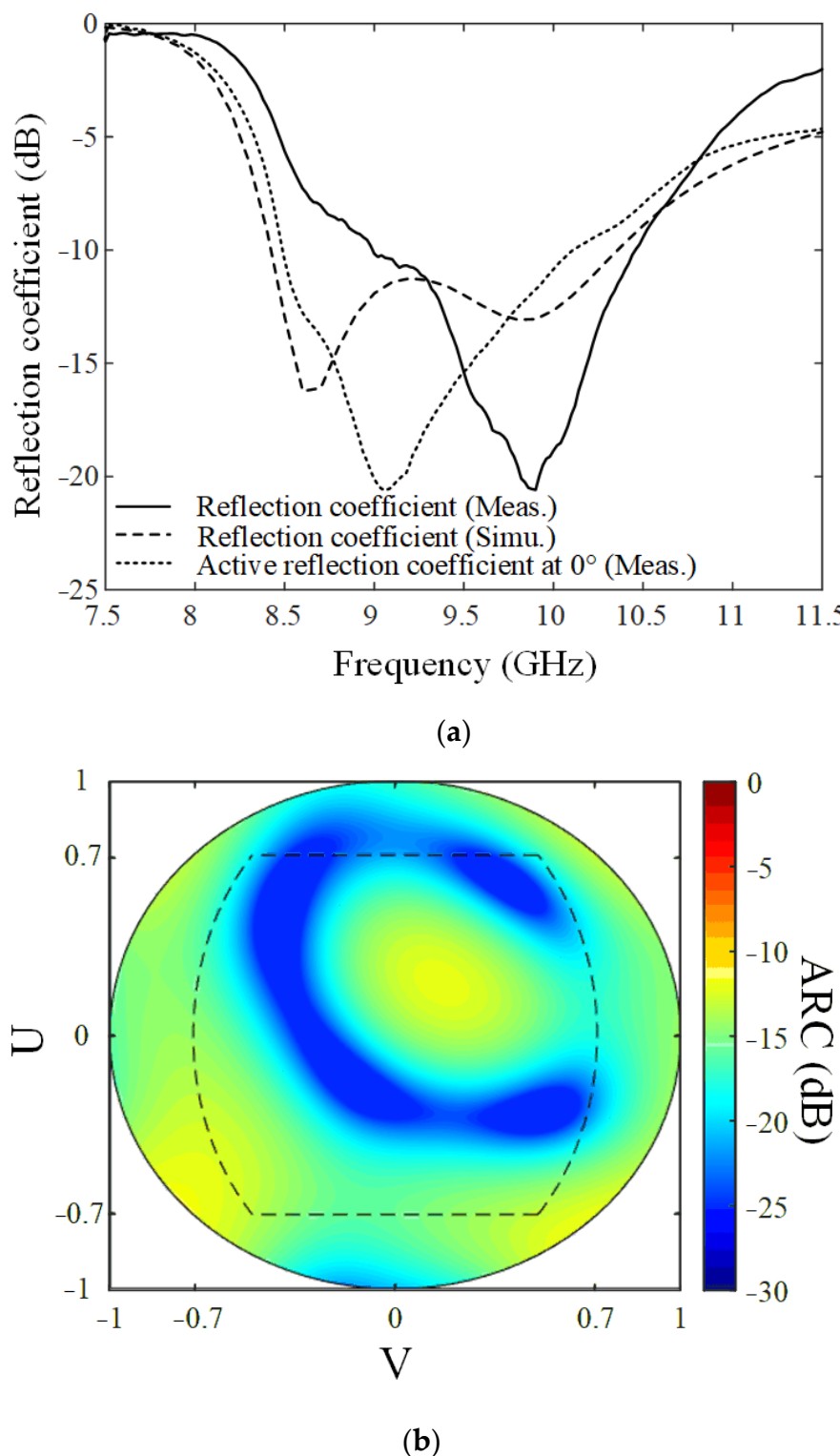

**Figure 6.** Active reflection co-efficient in X-band: (**a**) comparison between the reflection co-efficients and the active reflection co-efficient; (**b**) U–V plot.

Due to its the symmetrical modularized geometry, the proposed array antenna can be easily expanded to the various array shapes with a large aperture. Figure 8 describes the geometry of the expanded 12 × 12 X-band array configuration, where the side length (*L*) of the square array is 204 mm. Then, the expanded array can be mounted on a ship, as shown in Figure 9a. Figure 9b illustrates the simulated radiation pattern of the expanded array, where

the blue and red lines denote the expanded array beam pattern before and after mounting on the ship mast, respectively. The bore-sight gain of array with ship mast is 35.6 dBi, which is about 0.8 dB higher than the bore-sight gain of array without ship mast. The side lobe level (SLL) for the array beam pattern is about −13.3 dB. Figure 9c,d presents the simulated radiation patterns for the expanded 4 × 4 S-band array. The bore-sight gain of the array with the ship mast is 23.249 dBi, which is about 0.998 dB lower than the bore-sight gain of array without the ship mast. The SLL for the array beam pattern is about −22 dB.

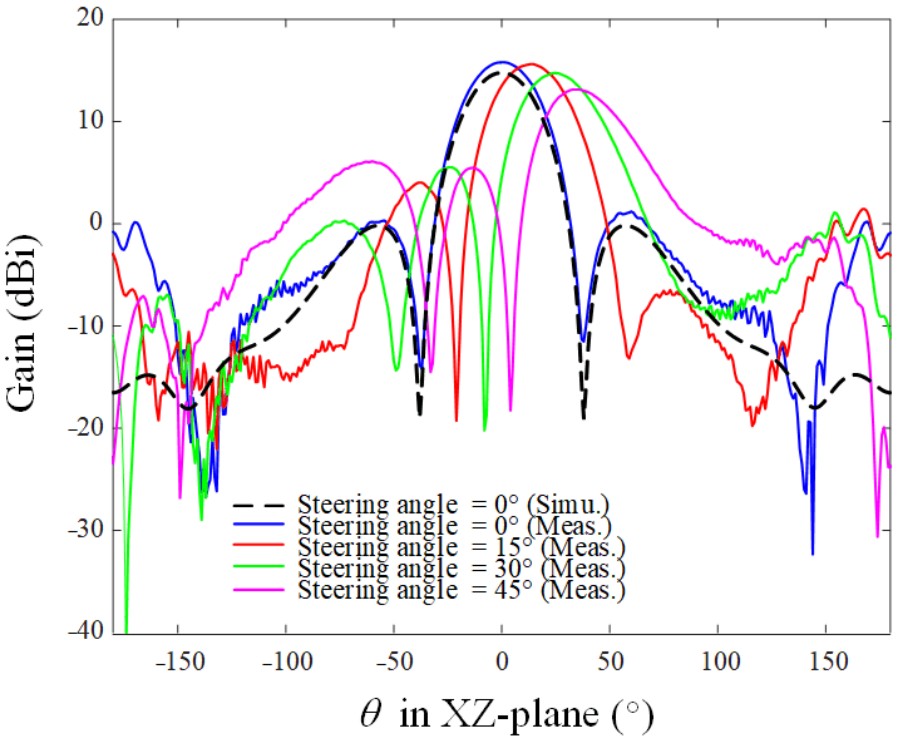

**Figure 7.** Simulated and measured steering array beam patterns at 9.5 GHz.

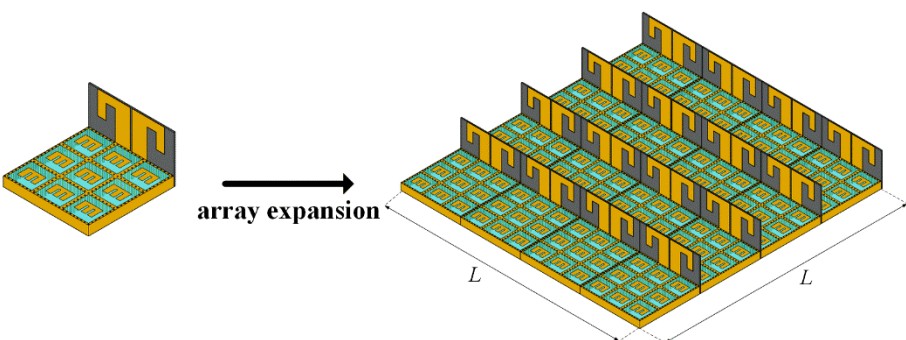

**Figure 8.** The geometry of the 12 × 12 X-band array configuration.

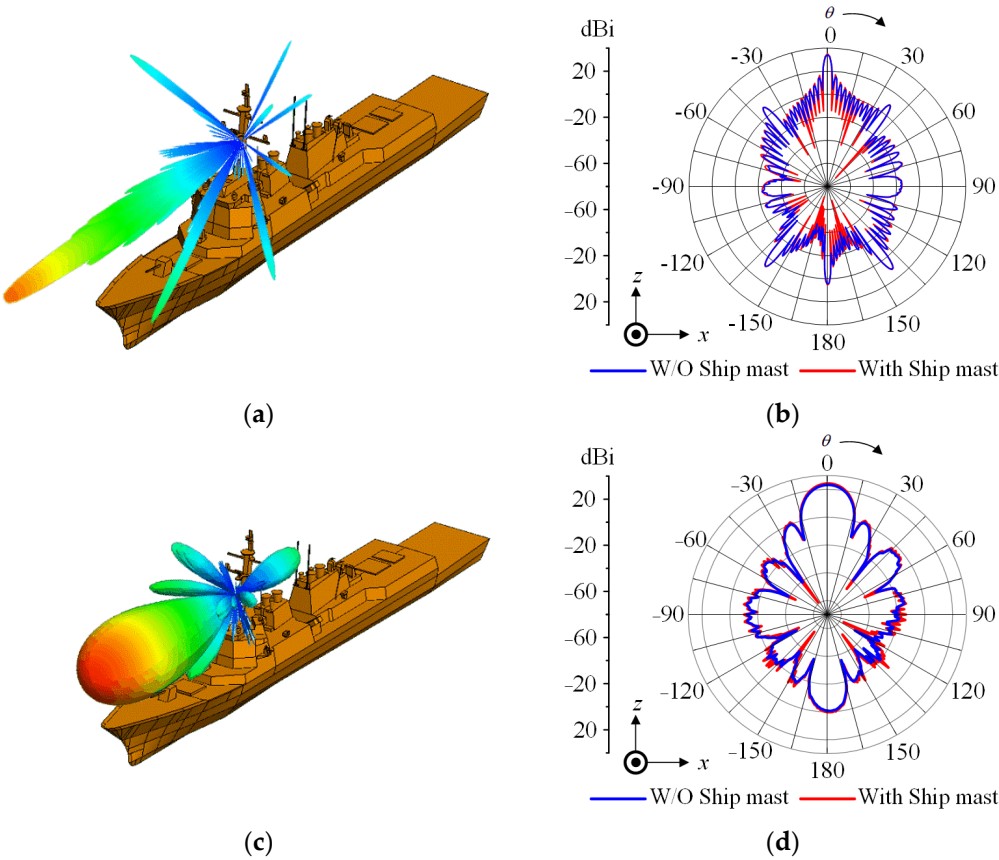

**Figure 9.** Simulated radiation patterns before and after mounting on the ship mast: (**a**) 3D radiation patterns after mounting on the ship mast in the X-band; (**b**) radiation patterns before and after mounting on the ship mast in the X-band; (**c**) 3D radiation patterns after mounting on the ship mast in the S-band; (**d**) radiation patterns before and after mounting on the ship mast in the S-band.

## 4. Conclusions

We proposed a multi-band array antenna that shared a common aperture with heterogenous array elements. The proposed array comprised one printed dipole antenna for S-band and nine E-shaped patch antennas for X-band. The current directions of the different band elements were orthogonal to each other, which properly decreased the mutual coupling interference. Moreover, to lower the mutual coupling interference among the X-band components, we installed cavities using multiple vias surrounding the X-band components. To validate the usefulness of the proposed design, the unit-cell was expanded to the 12 × 12 X-band array configuration, and then the beam steering properties were observed in the X-band. The proposed antenna's bandwidth and average gain were 890 MHz and 5.2 dBi in the S-band and 1.51 GHz and 5.2 dBi in the X-band. The bore-sight gain of the extended array configuration after mounting on the ship mast was 35.6 dBi. The results confirmed that the proposed design was applicable for MFR applications even in a shared aperture.

**Author Contributions:** Conceptualization, S.W., H.K. and H.C.; methodology, S.W., H.K. and D.K.; software, S.W. and H.K.; validation, S.W., H.K., H.C. and D.K.; formal analysis, S.W.; investigation, S.W.; resources, S.W. and H.K.; data curation, H.K. and S.W.; writing—original draft preparation, S.W.; writing—review and editing, S.W. and H.C.; visualization, S.W. and H.K.; supervision, H.C.; project administration, D.K.; funding acquisition, D.K. All authors have read and agreed to the published version of the manuscript.

**Funding:** This research received no external funding.

**Institutional Review Board Statement:** Not applicable.

**Informed Consent Statement:** Not applicable.

**Data Availability Statement:** Not applicable.

**Acknowledgments:** This research has been supported by the Challenging Future Defense Technology Research and Development Program (9127786) of Agency for Defense Development in 2019.

**Conflicts of Interest:** The authors declare no conflict of interest.

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
