# Peer review of "Multi-Band Array Antenna Sharing a Common Aperture with Heterogeneous Array Elements"

_applsci, doi:10.3390/app12189348_

Round 1

Reviewer 1 Report

The proposed shared aperture for two distinctive frequency bands has a novelty in terms of its structure and meaningful for real-world applications.  Here are some comments to improve the clarity of the manuscript.

1. Please include the substrate thickness and loss tangent for the S-band antenna (is it the same to the X-band antenna substrate?).

2. Please describe more clearly about the antenna combination. Are the S- and X-band antennas sharing the ground? Are the X-band cavity and S-band antenna electrically connected or floating to each other?

3. Figure 4(b) seems to be the gain of a single X-band antenna. If so, please state it in the text or caption. 

4. Please describe how the beam steering performance was measured (Figure 7). How was the phase controlled for each element? What is the steering direction (XZ-plane? YZ-plane?). Any impact in X-band beam steering due to the S-band antenna blockage?   

Reviewer 2 Report

The paper presents a heterogeneous array of microstrip patch antennas for multi-band radar applications. Orthogonality between elements of different type is exploited to reduce mutual coupling effects, and the steering performance in the 0-45 degrees range are checked through numerical and experimental validations.

The design represents a useful engineering application and hence, in the referee's opinion, the idea itself deserves publication. However, some technical and presentation issues are present. Hence, a major revision is required to reinforce the scientific soundness of the submitted study. The main issues found by this referee are reported in the following of this review.

1) The motivations for the selected design is unclear. The authors should better explain the reasons of their choices in the Introduction together with the contribution with respect to the existing design approaches.

2) Concerning steerable patch antenna arrays with beam reconfiguration, many other studies have been presented in further fields (5G, satellite, drones, ...) , such as, for example: (i)  “Reconfigurable phased antenna array for extending cubesat operations to Ka-band: Design and feasibility,” Acta Astronaut. 2017; (ii) “Orthogonal Printed Microstrip Antenna Arrays for 5G Millimeter-Wave Applications,” Energies, 2022; (iii) “Compact Switched-Beam Array Antenna with a Butler Matrix and a Folded Ground Structure,” Electronics, 2020; etc. These studies should be considered in the literature overview and discussed in comparison with the proposed design.

3) A discussion on the polarization supported by the proposed array should be inserted in the results.

4) The meaning and the evaluation of the s_{mn} parameter in (1) should be explained together with a description of the feeding network.

Reviewer 3 Report

These are my comments:

1. The paper is very  interesting, but I think there is many important information that was not included into the paper.

2. As the array is multiband, the paper should include parameters of both resonance frequencies (3.25 GHz and 9.75 GHz). For instance, it should also include the ARC for the first frequency 3.25 GHz. The steering array beam pattern at 3.25 GHz is an important data to know.

3. The Figure 9b is at 9.5 GHz?, how is the pattern with ship mast at 3.25 GHz?

4. What is the material of the ship mast?

5. How long does the simulation last with the ship mast?

6. What is the math model of the unit cell?

7. The Figure 1c shows a sma connector for each E element, does the unit cell contain nine sma connectors for X band operation?

8. What is the material of the antennas?

9. What is the SLL vale of the Figure 9b?

10. Is the gain of Figure 4b of one E antenna element? is it the center E antenna element?

11. Include all the reflection coefficients of the nine E antenna elements.

12. How is the ARC at 9.75 GHz and 3.25 GHz when the beam is steered at 45 degrees?

13. How are the steering patterns at 3.25 GHz?

14.  How is the 3D pattern at the 3.25 GHz for  0 degrees?

15. Include a picture of the back side prototype. This is important to observe how are the sma connectores.

16. Include a Table with the comparison of the proposed design with respect to antenna arrays of the literature. The Table can include frequency, size, number of antennas, technology, gain and steering capabilities. You can include examples of the references [10], [11] and [12].

17. The proposed array example is 4 x 4 unit cells. Why you mentioned it is 12 x 12 array?

18. The proposed array extension considers 160 sma connectors? 10 sma connectors for each unit cell. Include a discussion of how would you recommend to design such complex beamforming network in future works.

Round 2

Reviewer 1 Report

The authors have properly replied to questions and comments. The quality of the manuscript has be improved, thus ready for publication. 

Reviewer 2 Report

The paper proposes the design of a heterogeneous antenna array of microstrip patches for multi-band radar equipments. Mutual coupling among elements is reduced by exploiting the orthogonality between elements of different type, while the steering capabilities in the 0-45 degrees range are investigated through simulations and measurements.

With respect to the previous version the background, the research content and the quality of the presentation has been improved. All issues signalled by this referee have been addressed, this, in the reviewer's opinion, the paper is suitable for publication.